# Differentially Private Image Classification Using Support Vector Machine and Differential Privacy

**Makhamisa Senekane** 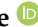

Department of Physics and Electronics, National University of Lesotho, P.O. Roma 180, Maseru 100, Lesotho; mc.senekane@nul.ls

**Abstract:** The ubiquity of data, including multi-media data such as images, enables easy mining and analysis of such data. However, such an analysis might involve the use of sensitive data such as medical records (including radiological images) and financial records. Privacy-preserving machine learning is an approach that is aimed at the analysis of such data in such a way that privacy is not compromised. There are various privacy-preserving data analysis approaches such as *k-anonymity*, *l-diversity*, *t-closeness* and Differential Privacy (DP). Currently, DP is a golden standard of privacy-preserving data analysis due to its robustness against background knowledge attacks. In this paper, we report a scheme for privacy-preserving image classification using Support Vector Machine (SVM) and DP. SVM is chosen as a classification algorithm because unlike variants of artificial neural networks, it converges to a global optimum. SVM kernels used are linear and Radial Basis Function (RBF), while $\epsilon$-differential privacy was the DP framework used. The proposed scheme achieved an accuracy of up to 98%. The results obtained underline the utility of using SVM and DP for privacy-preserving image classification.

**Keywords:** support vector machine; differential privacy; privacy; privacy-preserving machine learning; Modified National Institute of Standards and Technology (MNIST); image classification

---

## 1. Introduction

With a massive increase in the collection and storage of personal data such as medical records, financial records and census data and web search histories, the concern for privacy has been exacerbated [1]. This has resulted in the adoption of privacy-preserving data analysis; which seeks to enable analysis of data while at the same time ensuring that privacy of individuals in the dataset is not compromised [2–5]. To date, different approaches have been used to realize privacy-preserving data analysis. These include:

- *k-anonymity*: this approach was proposed by Sweeney in 2002 [6]. A dataset is said to be *k-anonymous* if every combination of identity-revealing characteristics occurs in at least *k* different rows of the dataset. This anonymization approach is vulnerable to such attacks as background knowledge attacks [7].
- *l-diversity*: it was proposed by Machanavajjhala et al. in 2007 [7]. The *l-diversity* scheme was proposed to handle some weaknesses in the *k-anonymity* scheme by promoting intra-group diversity of sensitive data within the anonymization scheme [8]. It is prone to skewness and similarity attacks [9].
- *t-closeness*: this anonymization scheme was proposed by Li et al. in 2007 [9]. It is a refinement of *l-diversity* discussed above [8]. It requires that distribution of sensitive attributes within each quasi-identifier group should be "close" to their distribution in the entire original dataset (that is, the distance between the two distributions should be no more than a threshold *t*) [9].

- *Differential Privacy (DP)*: It was proposed by Dwork et al. in 2006 [10]. Unlike anonymization schemes discussed above, DP provides information-theoretic guarantee that the participation of the individual(s) in a statistical database would not be revealed. It has since become a gold standard of privacy-preserving data analysis.

Machine learning is the most advanced field of artificial intelligence [11–14]. It enables the computers to learn from data without being explicitly programed. Machine learning can be broadly divided into three paradigms, namely [12]:

- Supervised learning: where the model uses both data and the corresponding labels. Supervised learning models/algorithms can be divided into classification and regression algorithms.
- Unsupervised learning: in this paradigm, there are no corresponding labels to data. The objective in this case is to generate the distribution that represent the data; hence why unsupervised learning models are also known as generative models.
- Reinforcement learning: this machine learning paradigm involves an agent which interacts with an environment, and gets either reward or penalty for the action taken while at a particular state. The ultimate purpose is for the agent to maximize cumulative reward.

In this paper, we propose a privacy-preserving image classification scheme using support vector machine (SVM) and DP. The image dataset used in the work reported in this paper is Modified National Institute of Standards and Technology (MNIST) dataset [15]. MNIST dataset is a dataset of handwritten digits that is used to train machine learning algorithms. Futhermore, two SVM kernels were used, namely linear and Radial Basis Function (RBF) kernels. Furthermore, $\epsilon$-DP definition was used to realize privacy-preserving machine learning. The key contribution of the work reported in this paper is the design of privacy-preserving image classification algorithm which makes use of SVM and $\epsilon$-DP. This makes it possible to use sensitive multimedia files like images for privacy-preserving data analysis.

The remainder of this paper is structured as follows. The next section provides background information on SVM, DP and privacy-preserving machine learning, as well as related work concerning privacy-preserving machine learning. This is followed by Section 3, which discusses the privacy-preserving image classification algorithm proposed in this paper. Furthermore, Section 4 provides the results obtained and analyzes such results. The last section concludes this paper.

## 2. Background Information

### 2.1. Support Vector Machine

Support Vector Machine is a supervised machine learning algorithm which can be used for both classification (including multi-class classification) or regression tasks [13]. It uses the concept of maximum margin as a technique to avoid misclassification. Furthermore, it relies on the concept of support vectors [13]. SVM first identifies the hyperplane where data would be linearly separated. Thus, if data boundary (separator) looks non-linear in one hyperplane, SVM uses a technique called "kernel trick" to transform data to higher dimension, where such data could be linearly separable. SVMs do converge to global optima.

Support vectors of an SVM are input features/vectors that just touch the boundary of the margin of separating hyperplane of Support Vector Machine. Such support vectors are given as [12]:

$$W_0^T \mathbf{X} + b_0 = +1 \tag{1}$$

or

$$W_0^T \mathbf{X} + b_0 = -1, \tag{2}$$

for a binary classification SVM, where $W_0$ are weights, $\mathbf{X}$ are input features, $b_0$ are biases, $T$ are matrix transposes, and $\pm 1$ are classification classes.

*2.2. Differential Privacy*

Differential privacy is the strongest definition of privacy which was first proposed by Dwork in 2006 [10,16–20]. It provides a guarantee that the participation of an individual or (a group of individuals) in a statistical database not compromise the privacy of such an individual (or a group of individuals) [17].

**Definition 1.** *A randomized function* K *gives an $\epsilon$-differential privacy if for all datasets $D_1$ and $D_2$ differing in at most one element, and all* S $\subseteq$ *Range(*K*) [10],*

$$\mathbb{P}[K(D_1) \in S] \leq exp(\epsilon) \times \mathbb{P}[K(D_2) \in S], \tag{3}$$

*where $\epsilon \ll 1$.*

**Definition 2.** *A randomized function* K *gives an $(\epsilon, \delta)$-differential privacy if for all datasets $D_1$ and $D_2$ differing in at most one element, and all* S $\subseteq$ *Range(*K*) [16],*

$$\mathbb{P}[K(D_1) \in S] \leq exp(\epsilon) \times \mathbb{P}[K(D_2) \in S] + \delta, \tag{4}$$

*where $\epsilon \ll 1$ and $\delta$ is cryptographically small.*

*2.3. Privacy-Preserving Machine Learning*

In recent years, there has been a considerable research interest in privacy-preserving machine learning [21–27]. This technique combines machine learning techniques (such as logistic regression, SVM, artificial neural networks, etc.) with privacy schemes such as Differential Privacy in order to enable gaining insights from data while at the same time maintaining the privacy of the individuals' data.

Chaudhuri and Hsu [21] explore the complexity bounds for the differentially private classification supervised learning. On the other hand, in Reference [24], Apple's Differential Privacy Team proposes a local Differential Privacy system that is both efficient and scalable. This system uses local Differential Privacy approach. Lastly, Uber also uses Differentially Privacy for end-to-end data analysis. The privacy-preserving scheme used by Uber is based on elastic sensitivity [26].

Choi et al. [27] propose a privacy-preserving scheme for application in Internet of Things (IoT) systems which use ultra-low power. This proposed scheme uses local Differential Privacy. In Reference [28], Abadi et al. report a privacy-preserving machine learning algorithm using deep learning (which is a variant of artificial neural networks) and Differential Privacy. As already stated, artificial neural networks tend to converge to local optima, as opposed to global optima. On the other hand, Chaudhuri and proposed a privacy-preserving Empirical Risk Minimization (ERM) schemes for both logistic regression and SVM [29]. Additionally, Rubinstein et al. proposed a privacy-preserving mechanism for text classification using SVM [30].

## 3. Privacy-Preserving Image Classification Algorithm

As already stated, the privacy-preserving image classification algorithm was implemented using SVM and $\epsilon$-Differential Privacy. The image dataset that was used for this classification is the MNIST dataset [15]. This dataset is informally referred to as the "fruit-fly" or the "hello world" of image processing because it is usually the first one to be used to test the utility of various image-based machine learning algorithms.

It consists of a database of hand-written digits (from 0 to 9). Figure 1 shows 100 images taken from MNIST dataset.

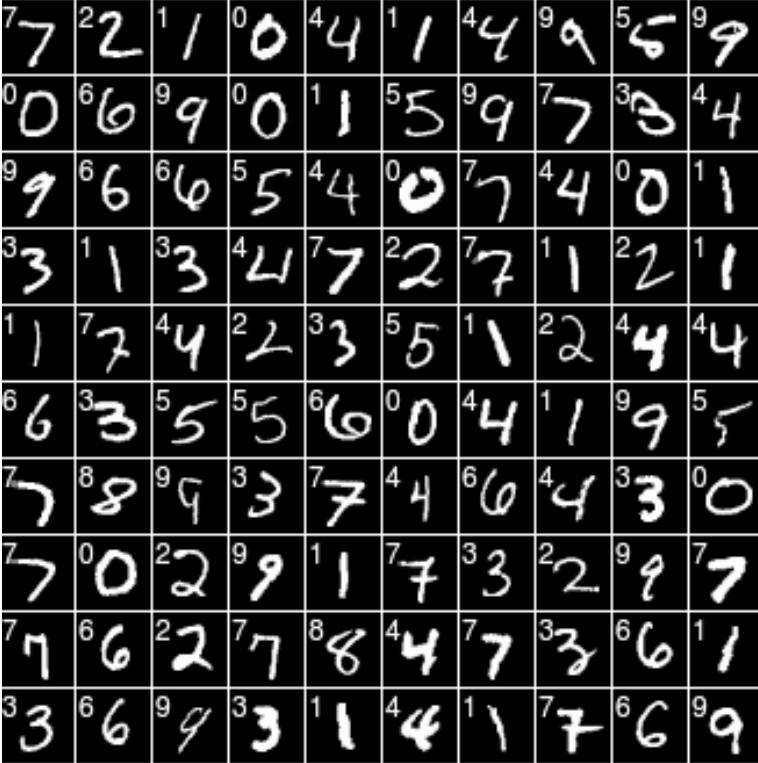

**Figure 1.** The 100 images from the Modified National Institute of Standards and Technology (MNIST) dataset. The image data is the central one, while the image label is given at the top-left corner of the image. Image courtesy of: http://eblearn.sourceforge.net/old/demos/mnist/index.shtml.

The programming language that was used to implement the proposed image classification algorithm is Python (www.python.org). Python was chosen because of its maturity and versatility. Machine learning framework that was chosen for this algorithm is Scikit-learn (www.scikit-learn.org). This is due to its popularity and simplicity. Furthermore, two SVM kernels; namely linear and RBF, were used in the work reported in this paper. In each case, the kernel was used first without noised training data and then with the noised training data. Then, using the pure (unnoised) testing data, the accuracies of these different approaches were determined. It is worth noting that the noise was added to each pixel of the images. Figure 2 shows pictures of digit 1 where it is unnoised, and where it is noised with various values of $\epsilon$ (privacy loss). This figure shows that higher $\epsilon$ implies lower privacy.

Finally, the image classification uses $\epsilon$-DP, where $\epsilon$ ranges from 0.01 to 10.

The algorithm for implementing the proposed scheme is summarized in Table 1.

**Table 1.** An algorithm for implementing privacy-preserving Modified National Institute of Standards and Technology (MNIST) classification.

---

**Algorithm 1** Privacy-preserving Image classification

---

**Start**
1: Load image dataset
2: Split dataset into train and test data
3: Add Laplace noise to the train data to privatize it
4: Train the model using Support Vector Machine (SVM) and noised data
5: Test the model using unnoised test data
**End**

---

**Figure 2.** MNISt digit 1 (**top left**). The same digit is noised with $\epsilon$=0.01 (**top right**), $\epsilon$=ln2 (**bottom left**), and $\epsilon$=10 (**bottom right**). As can be observed from this figure, low value of $\epsilon$ (privacy loss) results in high privacy.

## 4. Results and Discussion

Model accuracy was used as a metric for measuring performance of the designed image classification models. Table 2 provides a summary of accuracies for different implementations of SVM kernels; both pure and noised (differentially private SVM), with $\epsilon$ =ln2. From the table, it can be observed that the performance of privacy-preserving image classification SVM is comparable to that of the pure image classification SVM. These results underline the utility of the proposed differentially private SVM (DP-SVM).

**Table 2.** Accuracies of different implementations of image classification SVMs. Differential privacy, DP; Support vector machine, SVM: Radial basis function, RBF.

| SVM | Linear Kernel Accuracy (%) | RBF Kernel Accuracy (%) |
|---|---|---|
| Pure | 97.8 | 98.6 |
| DP-SVM | 97.2 | 98.3 |

Confusion matrices for different implementations of MNIST image classification support vector machines with $\epsilon$ =ln2 are provided in Table 3. Once again, these results demonstrate that the performance of pure SVM is comparable to that of DP-SVM, since their misclassification errors are comparable (and low). Therefore, these results also underline the utility of the privacy-preserving image classification scheme proposed in this paper.

**Table 3.** Confusion matrices for various implementations of image classification SVMs.

| SVM | Linear Kernel Confusion Matrix | RBF Kernel Confusion Matrix |
|---|---|---|
| pure | [[33 0 0 0 0 0 0 0 0 0]<br>[ 0 28 0 0 0 0 0 0 0 0]<br>[ 0 0 33 0 0 0 0 0 0 0]<br>[ 0 0 0 32 0 1 0 0 0 1]<br>[ 0 1 0 0 45 0 0 0 0 0]<br>[ 0 0 0 0 0 47 0 0 0 0]<br>[ 0 0 0 0 0 0 35 0 0 0]<br>[ 0 0 0 0 0 0 0 33 0 1]<br>[ 0 0 0 0 0 1 0 0 29 0]<br>[ 0 0 0 1 1 0 0 1 0 37]] | [[33 0 0 0 0 0 0 0 0 0]<br>[ 0 28 0 0 0 0 0 0 0 0]<br>[ 0 0 33 0 0 0 0 0 0 0]<br>[ 0 0 0 33 0 1 0 0 0 0]<br>[ 0 0 0 0 46 0 0 0 0 0]<br>[ 0 0 0 0 0 46 1 0 0 0]<br>[ 0 0 0 0 0 0 35 0 0 0]<br>[ 0 0 0 0 0 0 0 33 0 1]<br>[ 0 0 0 0 0 1 0 0 29 0]<br>[ 0 0 0 0 0 0 0 1 0 39]] |
| DP-SVM | [[33 0 0 0 0 0 0 0 0 0]<br>[ 0 28 0 0 0 0 0 0 0 0]<br>[ 0 0 32 0 0 0 0 0 1 0]<br>[ 0 0 0 33 0 1 0 0 0 0]<br>[ 0 1 0 0 45 0 0 0 0 0]<br>[ 0 0 0 1 0 46 0 0 0 0]<br>[ 0 0 0 0 0 0 35 0 0 0]<br>[ 0 0 0 1 0 0 0 32 0 1]<br>[ 0 0 0 0 0 1 0 0 29 0]<br>[ 0 0 0 1 0 0 0 1 1 37]] | [[33 0 0 0 0 0 0 0 0 0]<br>[ 0 28 0 0 0 0 0 0 0 0]<br>[ 0 0 33 0 0 0 0 0 0 0]<br>[ 0 0 0 33 0 1 0 0 0 0]<br>[ 0 0 0 0 46 0 0 0 0 0]<br>[ 0 0 0 0 0 47 0 0 0 0]<br>[ 0 0 0 0 0 0 35 0 0 0]<br>[ 0 0 0 1 0 0 0 32 0 1]<br>[ 0 0 0 0 0 1 0 0 28 1]<br>[ 0 0 0 0 0 0 0 1 0 39]] |

In order to investigate the privacy-utility tradeoff of the proposed algorithm, different values of $\epsilon$ were used for both the linear SVM and the RBF SVM. Table 4 gives the summary of the accuracies for different values of $\epsilon$.

**Table 4.** Privacy-utility tradeoff for different values of $\epsilon$.

| $\epsilon$ | Linear SVM Accuracy (%) | RBF SVM Accuracy (%) |
|---|---|---|
| Pure SVM | 98.1 | 98.6 |
| 0.01 | 1.57 | 7.78 |
| 0.1 | 37.6 | 43.9 |
| ln2 | 97.2 | 98.3 |
| 5 | 97.8 | 98.5 |
| 8 | 98.1 | 98.6 |
| 10 | 98.1 | 98.6 |

The work reported in this paper was limited to only one image dataset, namely MNIST dataset. Future work will explore another image dataset which is gradually proving to be yet another "fruit-fly" of image processing. This dataset is called fashion-MNIST (www.github.com/zalandoresearch/fashion-mnist).

## 5. Conclusions

We have reported the privacy-preserving image classification scheme which uses support vector machine and $\epsilon$-Differential Privacy. This scheme achieved the accuracy of up to 98% on test data. The high test accuracy of the scheme, which is comparable to the pure SVM, confirms the utility of the proposed scheme for image classification. Future work will explore the use of fashion-MNIST dataset for privacy-preserving image classification. The ultimate goal is to use real-life dataset for privacy-preserving image classification using SVM and Differential Privacy.

**Funding:** The author thanks National University of Lesotho for funding.

**Acknowledgments:** The author acknowledges the support of National University of Lesotho Research and Innovations Committee.

**Conflicts of Interest:** The author declares no conflict of interest.

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
