# Peer review of "Differentially Private Image Classification Using Support Vector Machine and Differential Privacy"

_make, doi:10.3390/make1010029_

Round 1
Reviewer 1 Report
Please address the following questions to strengthen your paper.
- Please describe in more detail how you privatize the image. Did you add the Laplace noise to each pixel independently? Could you also add some figures showing how the image looks like after privatization (i.e. adding noise)?
- The way that you implemented and trained DP+SVM is already described in the following paper:
W.-S. Choi, et al. "Guaranteeing local differential privacy on ultra-low-power systems" in ISCA 2018. (section 6-F)
The difference is that you tested on MNIST with a different kernel (RBF). (Also they used some modified DP mechanism.) I think that readers would find your paper more interesting if you can show some tradeoff between privacy and utility in the result section like the way the upper paper did.
Since typically when DP is used for privacy-preserving machine learning, higher privacy level requires more training data (or time) to obtain the same level of accuracy. Thus showing this tradeoff for your experiment (by sweeping the privacy parameter epsilon) would provide more information to the readers.
Author Response
Response to the Reviewer's Comments
Did you add the Laplace noise to each pixel independently? Could you also add some figures showing how the image looks like after privatization (i.e. adding noise)?
This comment has been addressed in the updated Manuscript. The noise was added to the individual pixel. Furthermore, Figures 2-5 show one of the MNIST digits, digit 1, before being noised and after being noised with different values of ε.
The way that you implemented and trained DP+SVM is already described in the following paper:
W.-S. Choi, et al. "Guaranteeing local differential privacy on ultra-low-power systems" in ISCA 2018. (section 6-F)
The difference is that you tested on MNIST with a different kernel (RBF). (Also they used some modified DP mechanism.) I think that readers would find your paper more interesting if you can show some tradeoff between privacy and utility in the result section like the way the upper paper did.
Since typically when DP is used for privacy-preserving machine learning, higher privacy level requires more training data (or time) to obtain the same level of accuracy. Thus showing this tradeoff for your experiment (by sweeping the privacy parameter epsilon) would provide more information to the readers.
The author thanks the Reviewer for suggesting the paper by Choi et. al., which also discusses the use of differential privacy using support vector machine. The suggestion by the Reviewer to show the tradeoff between privacy and utility has been incorporated in the updated Manuscript, and the summary of that is given in Table 4. The author once again greatly thanks the Reviewer for bringing this Choi et. al. paper to their attention.
Reviewer 2 Report
Good paper. Accept.
Author Response
Good paper. Accept.
The author greatly thanks the Reviewer for this kind review.
Reviewer 3 Report
In this article, an image classification scheme that uses SVM and privacy vector machine and DP is presented.
However, another important performance index is the efficiency of the proposed method. The author should show the efficiency of the images classification method.
Author Response
p { margin-bottom: 0.1in; direction: ltr; color: rgb(0, 0, 10); line-height: 120%; text-align: left; }p.western { font-family: "Liberation Serif",serif; font-size: 12pt; }p.cjk { font-family: "Noto Sans CJK SC Regular"; font-size: 12pt; }p.ctl { font-family: "FreeSans"; font-size: 12pt; }In this article, an image classification scheme that uses SVM and privacy vector machine and DP is presented.
However, another important performance index is the efficiency of the proposed method. The author should show the efficiency of the images classification method.
The author thanks the Reviewer for the review. In the updated Manuscript, the efficiency of the proposed method is demonstrated in Table 4, which discusses the privacy-utility tradeoff, for different values of ε.
Round 2
Reviewer 1 Report
Thank you for addressing my comments.
I have some minor comments as follows:
- It seems that the captions of Fig.3-5 are wrong. Smaller epsilon provides better privacy, so the values of epsilon should be in the opposite order.
- I think that it would look better if you make Fig.2-5 smaller and put them in one figure as subfigures.
- The readers would be more interested if you can add more explanation and references in the background Sec.2.3. Since your work privatizes the data using differential privacy, I think that following papers are relevant.
- U. Erlingsson, et al., "RAPPOR: Randomized Aggregatable Privacy-Preserving Ordinal Response," ACM CCS, 2014.
- Apple, "Learning with privacy at scale," Apple Machine Learning Journal, 2017
- K. Chaudhuri and D. Hsu, "Sample complexity bounds for differentially private learning," COLT, 2011.
- W.-S. Choi, et al., "Guaranteeing local differential privacy on ultra-low-power systems," ISCA, 2018.
Author Response
- It seems that the captions of Fig.3-5 are wrong. Smaller epsilon provides better privacy, so the values of epsilon should be in the opposite order.
This has been corrected, and the correct order has been provided. The author thanks the Reviewer for this insightful comment.
- I think that it would look better if you make Fig.2-5 smaller and put them in one figure as subfigures.
This suggestion has been incorporated in the updated Manuscript.
- The readers would be more interested if you can add more explanation and references in the background Sec.2.3. Since your work privatizes the data using differential privacy, I think that following papers are relevant.
- U. Erlingsson, et al., "RAPPOR: Randomized Aggregatable Privacy-Preserving Ordinal Response," ACM CCS, 2014.
- Apple, "Learning with privacy at scale," Apple Machine Learning Journal, 2017
- K. Chaudhuri and D. Hsu, "Sample complexity bounds for differentially private learning," COLT, 2011.
- W.-S. Choi, et al., "Guaranteeing local differential privacy on ultra-low-power systems," ISCA, 2018.
The suggested references (and explanations) were included in the updated Manuscript. The author thanks the Reviewer for these suggestions, which substantially improve the readability and comprehensibility of this Manuscript.
Reviewer 3 Report
In this article, an image classification scheme that uses SVM and privacy vector machine and DP is presented.
Author Response
p { margin-bottom: 0.1in; direction: ltr; color: rgb(0, 0, 10); line-height: 120%; text-align: left; }p.western { font-family: "Liberation Serif",serif; font-size: 12pt; }p.cjk { font-family: "Noto Sans CJK SC Regular"; font-size: 12pt; }p.ctl { font-family: "FreeSans"; font-size: 12pt; }a:link { }In this article, an image classification scheme that uses SVM and privacy vector machine and DP is presented.
The author thanks the reviewer for the comments. In the updated Manuscript, all the English language issues were addressed, to the best of the author’s capabilities.